# Embedding File Structure for Tabular File Preparation

## Abstract

We introduce the notion of file *structure*, the set of characters within a file's content that do not belong to data values. Data preparation can be considered as a pipeline of heterogeneous steps with the common theme of wrangling the structure of a file to access its payload in a downstream task. We claim that solving typical data preparation tasks benefits from an explicit representation of file structure. We propose a novel approach for learning such a representation, which we call a "structural embedding", using the raw file content as input. Our approach is based on a novel neural network architecture, composed of a transformer module and a convolutional module, trained in a self-supervised fashion on almost 1M public data files to learn structural embeddings. We demonstrate the usefulness of structural embeddings in several steps of a data preparation pipeline: data loading, row classification, and column type annotation. For these tasks, we show that our approach obtains performances comparable with state-of-the-art baselines on six real-world datasets, and, more importantly, we improve upon such baselines by combining them with the structural embeddings provided by our approach.

## 1 Data files = Payload + Structure

Plain text, CSV-like, tabular files are often used to create, store, distribute, and consume data. We refer to tabular content within these files as the *payload*, which is parsed from the raw file input. However, files often contain more than just characters representing table values: we define all characters within a data file that do not constitute its payload as the *structure* of a file. Even though a standard for the structure of CSV files exists (Shafranovich, 2005), real-world files often diverge from it and require significant effort to load correctly (Mitlöhner et al., 2016; Vitagliano et al., 2023; Hulsebos et al., 2023).

Consider the sample raw file of Figure 1: its payload is a table with three columns and three rows, but the structure of this file does not follow any official standard. Trying to automatically parse its table using existing algorithms would most likely lead to incorrect results. Given the complexity of data pipelines, rather than manually configuring each different system to load each different file optimally (if and as much as possible), users typically have to transform files into a standard version that is compatible with all the tools and processes required. We refer to this process, which is cumbersome, often manual, and time-consuming, as *file preparation*. With automated preparation far from being a reality (Kumar, 2021), each of these potential problems requires user effort to address, which notably makes up most of the practitioners' development time.

**Motivation:** Our motivation is to automate file preparation as much as possible, to reduce its burden on data scientists. We argue that all file preparation steps share the over-arching goal of understanding and transforming the structure of a file so that its payload can be parsed correctly. Often, data scientists are required to address these and other challenges individually, applying several tools and solutions. Leveraging general-purpose machine learning solutions to solve these tasks is challenging, due to the nature of data preparation problems: they are often tiny but with unique characteristics ("death by a thousand cuts"); high-quality labeled datasets to train on are expensive and rare to obtain; and typical general-purpose pre-trained models are designed for natural language text rather than tabular or numeric data (Rogers et al., 2020; OpenAI, 2023; Roziere et al., 2023).

**Problem statement:** Given a relational table T as a sequence of tuples that contain cell values for a given set of attributes, and given a file structure S as the set of characters within a tabular file that

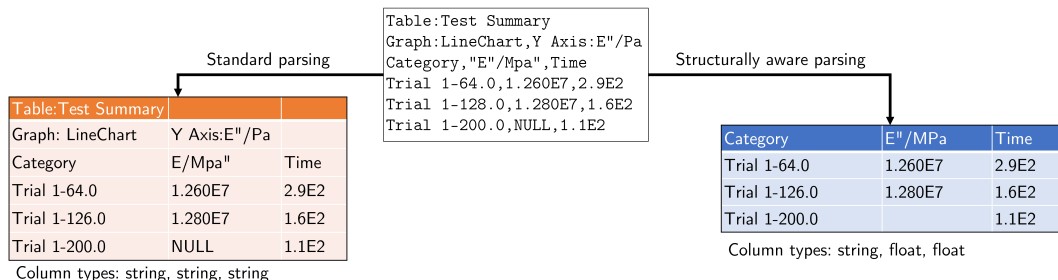

Figure 1: Parsing a raw tabular file correctly requires structural understanding.

do not map to either one of the table attributes (its "header") or values (its "cells"), the structural preparation problem can be expressed as:

*Given a tabular data file F serializing a table T with structure S, automatically and correctly detect its structure S to parse the attributes and tuples of its relational table T.*

We can further refine this problem into three specific subproblems:

1. **Dialect detection:** Given a file F, detect the structural characters of S to parse its rows.

2. **Row classification:** Given the rows of F, identify the tuples of its table T.

3. **Column type detection:** Given the tuples of T, identify meaningful attribute names.

**Intuition:** To overcome these challenges, our goal is to design a unique, task-independent model to represent file structure, specifically pre-trained on a large and structurally diverse set of files. With such a general model, file structure can be represented in vectorial embeddings that can be either fine-tuned for data preparation with smaller high-quality datasets or used as external features to enrich other specialized models. Inspired by the success of representation learning and pre-trained models in fields like natural language processing and computer vision, we propose RENEMB, a framework to encode cell-level, row-level, and file-level structure of tabular files as vectorial embeddings. The motivation for a specialized architecture stems from the fact that existing approaches for representation learning, including general-purpose LLMs (OpenAI, 2023; Roziere et al., 2023) and specialized tabular data models (Yang et al., 2022; Sun et al., 2023), focus on semantic tasks operating on the payload of files, and therefore assume correct parsing of file structure. In this paper, we experiment with such models (also see Appendix A) but found them inadequate to address structural preparation tasks.

**Paper contributions:** The main contribution of our work is the design of RENEMB, a novel framework to encode the structure of tabular text data files, leveraging transfomer- and convolution-based layers to address the file preparation problem. Our contribution includes a novel pattern tokenization strategy and two novel pre-training tasks, *structural masking* and *same file prediction*. Our experimental results demonstrate the effectiveness of these strategies for file preparation, compared to general-purpose LLMs that use natural-language or code oriented tokenization and training tasks (OpenAI, 2023; Roziere et al., 2023). We train this large model in a self-supervised fashion, leveraging the almost 1M real-world tabular files from the GitTables corpus (Hulsebos et al., 2023).

Furthermore, we contribute several strategies to apply the main RENEMB model and experiment with them on three preparation tasks: dialect detection, table understanding, and column type annotation. Our experiments demonstrate that thanks to the structural pre-training stage, RENEMB can be fine-tuned on these preparation tasks using relatively small labeled datasets and outperform other pretrained large models. We share all datasets with their annotations, the trained models with their weights, and the source code to reproduce the experiments[1].

---

[1]https://anonymous.4open.science/r/renemb-6E46/

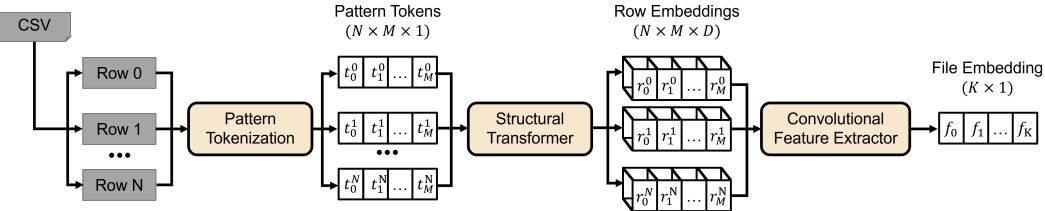

Figure 2: Three-tiered architecture of RENEMB.

## 2 THE RENEMB ARCHITECTURE

The architecture of RENEMB leverages three components, aimed at representing three levels of a tabular data file: *cells* with their sequence of characters, *rows* with their sequence of cells, and *files* with their sequences of rows. This architecture does not require any previous knowledge about a file's dialect to identify tabular cells and rows but rather uses self-supervised learning to understand its structure. To make this possible, the first step is what we call "pattern tokenization": this step produces a fixed-length sequence of tokens for every row. In Figure 2, individual tokens are named $t_0^0$ to $t_M^N$, where $N$ stands for the number of file rows, and $M$ for the length of the token sequences.

The next component is a transformer architecture based on BERT (Devlin et al., 2019) to encode tokens and file rows in vectors of dimension $D$ (set to 768 in our experiments). We pre-train this architecture on structural patterns rather than on natural language sentences. For the model to effectively capture structural rather than semantic features, we designed two novel pre-training tasks to encode the structure of every token and row. The final component of RENEMB aims at providing a single embedding, of dimension $K$ (set to 128), to capture file-wise structural features. The intuition behind the use of convolutional layers is their capability to capture spatial features and local structures. The remainder of this section explains in further detail each of these components.

### 2.1 PATTERN TOKENIZATION

Pattern tokenization aims at abstracting away the semantic information about the payload of a file, forcing the model to focus on structural properties. This is done by tokenizing the raw character stream of a file into what we call "structural patterns", that explicitly assign tokens to special characters and abstract cell values into regex-like strings. Figure 3 presents an example of such tokenization. The procedure to tokenize is, first, to split the character stream of a file into rows according to newline characters. For every row, we tokenize it according to all the special characters that it contains, while we encode everything in between two special characters with a "pattern". We define a "pattern" to encode one or subsequent alphanumeric characters. To encode patterns, we refer to Unicode character classes, that support multiple languages, including pictogram-based ones.

- A single lowercase letter or ideogram is represented as "l", contiguous sequences as "l*".
- A single uppercase letter is represented as "L", contiguous sequences as "L*".
- A single digit is represented as "d", contiguous digits as "d*".
- A pictogram or non-syntactic symbol is represented as "S", contiguous symbols as "S*".
- Contiguous strings of lowercase, uppercase, and symbols are represented as "T" (for text).
- Contiguous strings of numbers and text are encoded as "A" (for alphanumeric).

We expect the tokenization of rows to be highly consistent within the same file. Related data preparation research demonstrated the solidity of this assumption, by leveraging row consistency to detect or repair erroneous rows (Qahtan et al., 2018; Hameed et al., 2022). For example, tabular columns containing a number with periods separated with a dot as a decimal delimiter would all be represented with a "d*.d*" pattern, irrespective of their value. Due to the different lengths of cell values, however, the resulting tokenization of rows may not always align across different records. As described in the next section, RENEMB compensates for this effect with the attention mechanism of the row embedding transformer. Overall, the dataset we use for pretraining (Hulsebos et al., 2023) contains 409 unique pattern tokens in its vocabulary.

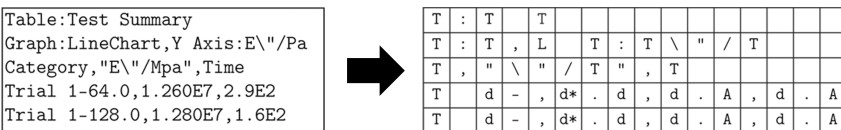

Figure 3: Pattern tokenization for the raw file of Figure 1.

## 2.2 STRUCTURAL TRANSFORMER

While pattern tokens are a representation of fine-grained cell structure, the sequence of values within rows embodies a further structural level of a tabular file. Learning to represent the structure of file rows can be thought of as a special "language modeling" task, since the composition of rows in data files also follows grammatical, syntactic, and semantic rules. Therefore, the intuition of the second component of RENEMB is to use a transformer neural network architecture, composed of 6 transformer encoder layers with 12 attention heads each. Adapting the tasks used in (Devlin et al., 2019), these structural transformers are pre-trained on *pairs* of file rows. The two input rows are tokenized using our pattern tokenization and concatenated with a [CLS] token at the beginning of the sequence, and a [SEP] token between the tokens of the two rows. Once pretrained, the [CLS] token vectors will embed a row-level representation. As pairs of file rows may have a different number of input tokens, we reserve $(M/2) - 1$ tokens for each row: rows with more tokens are truncated, while shorter ones are padded with the padding token [PAD]. Padding tokens are excluded from the attention calculations. After exploratory analysis of the pre-training dataset, we set $M$ to 128, a dimension that covers over 90% of all tokenized rows from the dataset files. The transformer pre-training is carried out on 10 million row pairs, extracted from the over 850 000 raw files from the GitTables corpus (Hulsebos et al., 2023). The transformer layers of RENEMB are pre-trained with two novel training tasks: "Structural Masking Modeling" and "Same File Prediction".

**Structural Masking Modeling:** The first objective is an adaptation of the general-purpose masked language modeling task from Devlin et al. (2019), which is defined for natural language sentences. Rather than masking all tokens in the patterns with equal probability, in structural masking modeling, we only mask special character tokens, regardless of their role within the file. To correctly solve the structural masking task, RENEMB has to learn which special characters belong within a cell, e.g., the comma in "2,331", and which mark its surroundings, e.g., double quotes and comma in the row 15,Response,"2,331",404. For this, the attention mechanism of transformer encoder layers plays a vital role: the context of a token within a row helps differentiate, for example, a comma that occurs within two quotation characters. Moreover, as the pre-training set includes a wide variety of files with different dialects, the model is discouraged from overfitting on a given dialect.

**Same File Prediction:** For the second pre-training objective, the model is trained to classify whether the two rows belong to the same file or not. To do so, the model uses a logistic regression classifier. The input to the classifier is the encoding of the [CLS] token for each row pair. We compute a loss function using binary cross entropy, and train the transformer layers with the sum of the same file prediction loss and the structure masking modeling loss. Rows from different files have different structural properties like the number of cells, dialect characters, or different data types. However, rows from different files with the same dialect or a similar schema would have a similar structure, while rows from the same file may also have a different structure (e.g., header vs. data rows). Therefore, by pre-training on the same file prediction task, RENEMB is forced to produce row-level encodings that represent meaningful structural aspects, e.g., the number of commas in a row that are actual column delimiters.

## 2.3 CONVOLUTIONAL FEATURE EXTRACTOR

Beyond the row level, there are structural features that pertain to the file level and not to individual rows: for example, the presence of several header rows or multiple tables (Christodoulakis et al., 2020). The third component of RENEMB, the convolutional feature extractor, aims at encoding file structure in a condensed embedding vector. To capture this, bi-dimensional locality plays an important role: portions of the file that are close together, either in the same column or row, are more likely to share structural properties.

To account for locality, RENEMB's file embedding component is a CNN autoencoder architecture inspired by DCGANs (Radford et al., 2016), composed of a ResNet-18 encoder (He et al., 2016) and decoder. The input to the encoder layers is the feature map obtained by stacking the row embeddings of the previous stage for all file rows. The final output of the encoder layers is an encoding of size 128 that represents the structural file embedding. We tuned the size of this embedding as a hyperparameter for our model, experimenting with dimensions in $[32, 64, 128, 256, 512]$. For pre-training the convolutional feature extractor, we freeze the structural transformer layers of RENEMB and generate row and token embeddings for the first 128 rows per file, using padding or truncation for files with fewer or more rows. As the training loss for the convolutional feature extractor, we use the Mean Squared Error (MSE) between the input feature map and the output feature map, excluding padding values. The convolutional feature extractor is pre-trained for 3 epochs on the full set of GitTables CSV files, reserving 10% of the files (87 139) for validation purposes.

## 3 DIALECT DETECTION

Dialect detection is required to parse tabular files that do not follow the CSV standard, which is not a rare occurrence (Vitagliano et al., 2023). Typically, heuristic or frequency-based algorithms are applied (Hübscher et al., 2023). The task is still challenging for several reasons: (1) algorithms are not designed for uncommon dialects; (2) file rows may have inconsistent dialects; (3) files may have "broken" dialects. To use RENEMB for the dialect detection task, we fine-tune it with a logistic classification head that takes as input the concatenation of each token-level embedding with the corresponding row and file embeddings, and outputs the probability for each token to be a cell value, delimiter, quotation, or escape character. As training loss, we use the cross-entropy losses calculated on the whole sequence of file tokens. The final dialect is chosen as the one corresponding to the tokens most frequently tagged as delimiter, quotation, and escape character. If, within a file, no token is classified as delimiter (or quote, or escape) we consider the file as having an "empty" delimiter (or quote, or escape).

**Experimental setup:** To perform fine-tuning, we use a training dataset of 18 300 CSV files, obtained with augmentation from an original set of 5 689 manually labeled files from van den Burg et al. (2019), unseen by RENEMB during pre-training. The augmentation was performed to ensure a balanced distribution of classes in the overall data. We describe in detail the augmentation process in Appendix B. We report the results on a validation set of 3 660 files (amounting to 20% of the training files), used for development purposes, and on an equally sized held-out test set, obtained using the same augmentation strategy but on a set of 1 543 held-out files never seen by RENEMB during both the pre-training and the fine-tuning phases. We trained RENEMB for 10 epochs using a batch size of 6. We evaluate the results of dialect detection using precision, recall and F1 score for each of the three dialect classes, as well as *dialect accuracy*, which is computed for each file by assigning a score of 1 only if all three classes (delimiter, quotation, and escape character) are correctly detected. The F1 scores are averaged across the different classes, weighting the average based on the number of samples for each class.

**Results:** To analyze the performances of RENEMB, we compared it with two baselines and three SOTA LLMs: CLEVERCSV, the state-of-the-art system for CSV parsing (van den Burg et al., 2019); a baseline obtained by fine-tuning XLM-RoBERTa, a pre-trained multilingual model (Conneau et al., 2019), with three classification heads, taking as input the [CLS] token embeddings for each file row, CHATGPT-3.5 in its version "turbo-1106"; CHATGPT-4 in its version "1106-preview"; and CODE-LLAMA in its "7B-Instructions" variant tuned for human coding instructions. We detail the chosen prompts in Appendix A.

Table 1 reports the results of our experimental evaluation, rescaling the scores to the range $[0, 100]$ for ease of comparison. As can be noted, RENEMB outperforms all baselines across all scores. We attribute the poor accuracy of the XLM-RoBERTa model to several factors. First, with a larger input size of 512 tokens, the model is significantly larger than RENEMB, whose input sequences are limited to 128. For this reason, it is very challenging to train the high amount of model parameters (561M) using a relatively small training dataset. This problem is further exacerbated by the nature of the model's pretraining – which included natural language text and not tabular file rows. Therefore, the attention layers of a pretrained language model are most likely focusing on semantic rather than on structural aspects of the file rows.

|  | Delimiter | | | Quote | | | Escape | | | Dialect |
|---|---|---|---|---|---|---|---|---|---|---|
|  | P | R | F1 | P | R | F1 | P | R | F1 | Acc. |
| CLEVERCSV | 79.01 | 52.43 | 60.58 | 70.68 | 51.23 | 56.89 | 23.50 | 20.52 | 21.91 | 33.68 |
| XLM-RoBERTa | 0.86 | 9.29 | 1.58 | 5.78 | 24.04 | 9.32 | 8.71 | 29.51 | 13.45 | 0.55 |
| CODE-LLAMA | 71.19 | 41.09 | 39.82 | 75.23 | 48.99 | 46.80 | 61.91 | 39.84 | 38.41 | 10.52 |
| CHATGPT-3.5 | 84.97 | 60.11 | 65.60 | 91.03 | 74.64 | 72.70 | 78.59 | 58.01 | 63.03 | 39.10 |
| CHATGPT-4 | 93.73 | 66.91 | 72.85 | 85.98 | 70.11 | 70.23 | 71.66 | 49.78 | 54.18 | 37.08 |
| RENEMB | **94.12** | **93.65** | **93.84** | **99.52** | **98.53** | **98.93** | **90.54** | **85.01** | **87.38** | **81.49** |

Table 1: Average precision, recall, F1 results for dialect detection on the test set (scaled to 0-100%).

Regarding the performance of CLEVERCSV, which leverages the notion of *row consistency* to identify the dialect resulting in a table with high column homogeneity, we noticed that it tends to overestimate the probability of a file having a standard dialect. This effect is probably caused by the weights for the dialect scoring in this approach being tuned using an imbalanced set of files with significantly more standard dialects, e.g., where over 80% of the files used comma as a delimiter.

Among the three LLMs considered, the CODE-LLAMA performances show the lowest overall dialect accuracy. In our experimental analysis, this model tended to overly classify files as having the most common dialect characters, e.g., comma or semicolon as delimiters and double quote as quotation characters, having a poor recall for files uncommon dialects, that were essentialy always detected as comma-delimited, double quote-quoted, and escaped with backslash. Interestingly, the two models from the GPT family have good precision for delimiter and quotation characters, but similarly show poor recall and a low overall accuracy. Manually analyzing the errors yielded by these two models, we identified several common mistakes: for many erroneous classifications, the model simply resorted to outputting the most common special character of the file, often also providing the same character as the delimiter, quotation, and escape character. These results show that, albeit these models show some promise in addressing the dialect detection task, they lack fine-grained understanding of file structure to solve it reliably and consistently, especially in difficult cases.

While analyzing the errors of RENEMB, we noticed how in its classification, it tended to perform worse on files with empty dialect characters, i.e., had no delimiter, quote, or escape. For example, the F1 score for the delimiter class restricted to files with no delimiter (i.e., single column files) is 74.57% compared to all other classes where the minimum F1 is 92.99%, for files delimited by space. Therefore, we can characterize RENEMB as favoring precision over recall, that is, it overestimates the probability that a file has an unusual dialect character over that no such character is present.

## 4 ROW CLASSIFICATION

Given a raw tabular file F, row classification aims at identifying the header and the data tuples of the table contained in F, and at the same time recognizing and extracting other useful metadata. This problem has been formulated in slightly different variations, we consider the same conceptual model and problem formulation used in the SOTA approach STRUDEL (Jiang et al., 2021), a random forest classifier. In this formulation, row classification is considered a multi-class classification problem where a row of a file can belong to one of the following mutually exclusive classes: *header*, *data*, *group*, *derived*, *metadata*, and *note*. The *header* rows contain the column names of a table; *data* rows represent the records of a table; *group* rows organize the table into sub-tables ("groups") and represent the header for a given group; *derived* rows contain data that is the result of some operation on *data* rows, e.g., a total, average, or aggregation; *metadata* and *note* rows contain metadata information respectively before and after a table. To utilize RENEMB for the row classification task, we fine-tune it with a shallow classification head consisting of two logistic layers. The first layer takes, for every row, all token-level embeddings and condenses them into a single dimension. The second layer combines the output of the first layer with the file-wise embedding computed by the convolutional encoder and outputs the class probabilities for each row.

**Experimental setup:** To perform row classification experiments, we leverage six publicly available datasets, introduced by Jiang et al. (2021) (See Appendix B), that are composed of generally unprepared real-world tabular files, i.e., files do not adhere to the CSV standard because they may contain non-data rows, multiline headers, or multiple tables. We train our model on the four datasets GOVUK, SAUS, CIUS, and DEEX, which contain a total of 1 162 files and 221 218 rows. Following

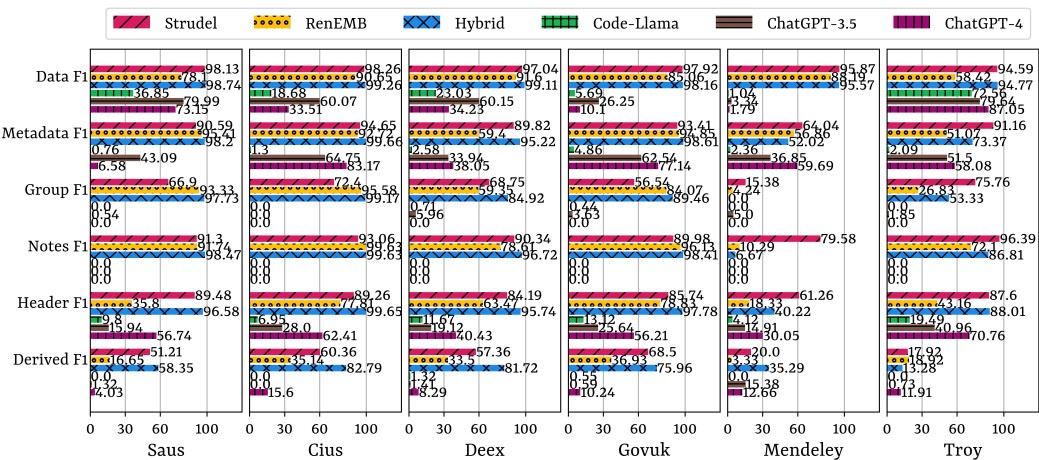

Figure 4: Results of row classification on the experimental datasets (scaled to 0-100%).

the experimental setup of Jiang et al. (2021), we train using 10 cross-validation folds on these four datasets. We also include two separate datasets, MENDELEY and TROY, for out-of-domain testing, containing 262 files and 199 946 rows. When testing on these two datasets, we used all files in the previous four datasets for training. We note that all files from the training or test sets have not been seen by RENEMB during pre-training. We train RENEMB for 40 epochs with a batch size of 8.

**Results:** We compare RENEMB with STRUDEL, the SOTA for row classification (Jiang et al., 2021). We also experimented with XLM-RoBERTa, using individual rows as input sequences and fine-tuning the model using the [CLS] embeddings to classify row classes. We tried different weighting and sampling strategies for the training set, however, in our experiments we were unable to train the model effectively towards classifying any row class apart from *data*, therefore we refrained from reporting its results. We attribute these challenges to the large dimension of the model compared to the relatively small and highly unbalanced training dataset (94.54% of the rows belong to the *data* class, see Appendix B), which speaks for the unique challenges of data preparation tasks. We also experimented with the three SOTA LLMs introduced in Section 3: CHATGPT-3.5, CHATGPT-4, and CODE-LLAMA using prompt engineering (see Appendix A).

Figure 4 reports the results of our experimental evaluation. The results highlight the competitive performances of RENEMB especially when classifying *note*, *metadata*, and *group* rows. These classes are often recognizable from their structure alone: *group* and *note* typically only have content in the first cells, and therefore are characterized by long sequences of delimiter characters. Furthermore, together with *metadata* rows, they are more likely to contain letter characters and fewer digits or numeric symbols. These structural features can be picked up by RENEMB thanks to its pattern tokenization strategy. In contrast, typically *data* and *derived* rows have a very regular structure. From our analysis of the results, the low scores of RENEMB for *data* and *derived* rows are due to the model frequently misclassifying one class as the other. This behavior is to be expected, considering that often the difference between *data* and *derived* rows lies within the content of their cells, a semantic detail that is abstracted away from RENEMB in favor of a structural perspective. To leverage the strengths of both models, we propose a HYBRID approach: we first run RENEMB to detect the class probabilities for the rows of a file, and then use these probabilities, one for each class, as extra features in STRUDEL. Since the features learned from RENEMB are biased towards structure, and those of STRUDEL are biased towards semantics, the HYBRID approach is an ensemble method to compensate for the bias of each individual model. This approach outperforms both STRUDEL and RENEMB with good success for every class in all cross-validation datasets.

However, as can be noted from the performances on the out-of-domain datasets, MENDELEY and TROY, the HYBRID approach does not lead to an improvement when any of the two base models has poor performances, or when the datasets have a very skewed distribution of *data* vs. non-*data* rows (Cf. Appendix B). This behavior is particularly evident, for example, for the *group* and *note* classes for MENDELEY.

| Dataset | Weighted-mean F1 | Macro-mean F1 |
|---|---|---|
| Unprepared | 77.96% | 66.55% |
| Auto-prepared with RENEMB | 80.50 % | 67.51 % |
| Manually prepared | 82.29 % | 71.06% |

Table 2: CTA performances with unprepared, automatically prepared, and manually prepared files.

None of the LLMs showed clearly superior performances over either RENEMB or STRUDEL, with all three models mostly struggling to detect *group* headers, *note*, and *derived* classes. Our experiments showed that in most cases, the models were classifying the first few lines of a file as either *metadata* or *header*, and all of the following lines as *data*, often demonstrating no true understanding of file structure. Moreover, the models often hallucinated in generating the answers for classification where many subsequent lines were identified for the same class, e.g., *data*. In these cases, if the row indices were used to indicate the classes (e.g., `data:  1,2,3,...`), the model answer turned to a naive enumeration of indices often even beyond the length of the file; if class labels were used as a response, e.g., `classes:  header, data, data,...`, after few repetitions for the same class the models kept repeating it over and over until answer was truncated. Although the most recent CHATGPT-4 shows improvements over CHATGPT-3.5, our experiments demonstrate that in their current stages, LLMs are unfit to reliably address the row classification task.

## 5    COLUMN TYPE ANNOTATION

Column type annotation (CTA) is defined in (Deng et al., 2020) as the task of annotating a column $c$ of a relational table $T$ with a semantic type $l \in L$ such that all values in $c$ belong to $l$. Successfully addressing the task for clean, standard relational tables is already challenging and the subject of wide research interest, given its importance for tasks such as data integration and discovery. We propose an end-to-end framework to perform CTA on raw tabular files, combining the structural embeddings of RENEMB with RECA, the SOTA model (Sun et al., 2023). First, for all files in the dataset, we obtain structural embeddings with the specialized RENEMB model for row classification described in the previous section. Then, we combine these embeddings with the features computed by STRUDEL and run our HYBRID row classification model. Using the row classes, we define a simple set of heuristics that can be easily automated to extract relational tables out of a dataset of raw files: 1. We delete empty rows and those classified as *metadata*, *note*, and *derived*; 2. In the case of multiple detected *header* rows in the file, we merge them into a single row, each value being separated by a whitespace character; 3. If there are multiple tables (defined as multiple stretches of *header* followed by *data* rows), we extract the first table; 4. If there is no *data* row, we exclude the file from our dataset; finally, we run the column type annotation model on the prepared versions of the dataset files. Considering the lower performances of XLM-RoBERTa, CHATGPT-3.5, CHATGPT-4, and CODE-LLAMA on row classification, we do not report on experiments with the row classes detected from these models.

**Experimental setup:** To assess the impact of RENEMB on column type annotation, we leverage the six datasets of real-world files introduced for the row classification task. We experiment with three versions of these datasets: an *unprepared* version, corresponding to the raw file input; an *auto-prepared* version, corresponding to automatic preparing of the files based on the row classes obtained with RENEMB; and a *manually prepared* version, corresponding to a CSV-standard version of the files, prepared with the same procedure but with the ground truth row classes. To perform column type annotation, we used the SOTA system RECA (Sun et al., 2023). Following the original implementation and experimental setup publicly available in the code repository of RECA, we split our datasets into a train/validation fold and a test fold, respectively composed of 90% and 10% of the original files. We trained the RECA model on the train/validation folds using 10-fold cross-validation for 20 epochs and tested on the test fold, repeating the test three times. Finally, we train and validate RECA on unprepared, automatically cleaned, and manually cleaned versions of the dataset and compare the results. Since this system is reported to outperform all previously proposed models (Hulsebos et al., 2019; Zhang et al., 2020; Suhara et al., 2022), which are also based on column-level embeddings, we expect our findings to hold true for the whole family of models.

**Results:** Table 2 reports the weighted and macro mean F1 score for all column type classes averaged across the three runs (standard deviations were zero). As can be noted, using *unprepared* versions

of the input files leads to the lowest performance. This is not surprising, considering that RECA encodes all the values that belong to a table column. If columns are parsed out of the unprepared file, they may contain (1) unrelated values; (2) values with heterogeneous types, e.g., if multiple tables or *derived* rows are contained in the file; or (3) be spuriously filled with empty values. On the contrary, automatically preprocessing the files with the row types detected thanks to RENEMB in the HYBRID scenario, improves the weighted F1 to 80.50%. To put in perspective the contribution of preparing the files with RENEMB, performing CTA on the *standard* files, prepared using manually annotated ground truth, leads to a further improved F1 of 82.29%.

## 6 RELATED WORK

Our work lies at the intersection of data management and representation learning. We briefly discuss how our work relates to the previous contributions in these areas.

**Table representation learning:** Recently, several works leveraged representation learning using transformer architectures or pre-trained language models on tabular data. Some tasks successfully addressed include column type annotation (Sun et al., 2023), error detection and data cleaning (Tang et al., 2021), and data discovery (Fan et al., 2023). An extensive review of the unique features of these models is beyond the scope of this work, and we refer readers to a recent survey (Badaro et al., 2023). One common aspect of all the aforementioned approaches that differentiates them from RENEMB, is that they are pre-trained on *relational tables* rather than on *tabular files*. For this reason, the representations learned are semantic rather than structural, as they encode column and row contents. While a unifying framework was proposed in Xie et al. (2022) to include semantic knowledge from structured data files within pre-existing language models to address table understanding tasks, it cannot be directly applied to *messy* files that require preparation and parsing of the payload table(s). Another line of related work has proposed representation learning frameworks to parse tabular data from visually rich documents (e.g., in PDF format) (Nassar et al., 2022; Li et al., 2021; Yu et al., 2023). These models employ a visual feature extraction stage (using either OCR or CNN methods) to identify tabular structures in images, and employ pretrained language models to understand their semantic content. In contrast, our work applies to text data and uses a transformer model trained to represent the structure of data files, while leveraging a convolutional stage to model locality within feature maps. The representations learned by RENEMB are orthogonal to the ones learned by the aforementioned tabular models, and can be compared with those learned by LLM instructed on code understanding (Roziere et al., 2023). We include these models in our experiments, and demonstrate that for data preparation tasks with few annotated labels, their performances are not competitive with a specialized model such as RENEMB.

**Algorithmic data preparation:** Often, algorithmic approaches proposed in the literature to solve file preparation tasks rely on structural features of the file such as row length, percentage of digit characters, etc., to train specialized machine learning models (Christodoulakis et al., 2020; Zhang et al., 2020; Döhmen et al., 2017). The main limitation of the feature engineering approach is the lack of generality, as it requires manually defining and possibly tuning the set of features during development with a given corpus of labeled data at hand. The approach we propose in this paper, instead, aims at learning a general representation of the structure of a file, which is task-independent and trained in a self-supervised fashion.

## 7 CONCLUSIONS

In this paper, we focus on the *structure* of tabular files, as opposed to their *payload*. We designed RENEMB, a novel neural network architecture to represent the structure of tabular files with cell-level, row-level, and file-level embeddings. We demonstrated how the embeddings generated by RENEMB can be leveraged to solve a variety of preparation tasks. Our experiments with pre-trained language models demonstrate that their focus on semantic features makes them unfit for use in data preparation tasks, where manually labeled datasets for fine-tuning may be expensive or challenging to obtain. Our vision is that specialized foundational models such as RENEMB, thanks to their specific exposure to large amounts of data files, can and will be used to assist users at all stages of the preparation pipeline, either to automate cumbersome operations or to empower software engineering and decision-making.

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

## A  DATA PREPARATION WITH LLMS?

With the advent of Large Language Models (LLMs) like those in the GPT family (OpenAI, 2023), recent research has experimented with the use of these models for traditional data wrangling/cleaning tasks, for example in Narayan et al. (2022). The intuition of this approach is to use a pre-existing LLM and perform zero-shot or few-shot inference to solve data management tasks. We experimented using three SOTA models, CHATGPT-3.5, CHATGPT-4, and CODE-LLAMA to sample their capabilities to solve structural tasks like dialect detection and row classification in CSV files.

To find the best prompt for the models, we queried the models to test dialect detection on a subsample of 100 files from the ones we use in Section 3 and to test row classification on a subsample of 120 files from the ones we use in Sections 4 and 5. The prompt used were the following:

- Dialect detection (CODE-LLAMA): *Identify the delimiter, quotation and escape characters of the following CSV file. Provide the output of the classification as a JSON containing the keys "delimiter", "quotation" and "escape".*

- Dialect detection (CHATGPT-3.5,CHATGPT-4): *Identify the delimiter, quotation and escape characters of the following file. Provide the output as a JSON.*

- Row classification (CODE-LLAMA): *Header lines represent the column names of tables; data lines represent records; group lines organize tables into sub-tables and are the header for a given group; derived lines contain the result of some operation on data lines; metadata and note lines contain metadata information respectively before and after tables. For the following CSV file, please classify each line as one of the following classes: header, data, group, derived, metadata or note. Provide the output of the classification as a JSON containing the key "predicted_classes" with the list of predicted row classes.*

- **Row classification (CHATGPT-3.5, CHATGPT-4)** :*Header lines represent the column names of tables; data lines represent records; group lines organize tables into sub-tables and are the header for a given group; derived lines contain the result of some operation on data lines; metadata and note lines contain metadata information respectively before and after tables. In the following CSV file, identify what lines are data, header, group header, metadata, note, and derived. Provide the output as a JSON containing a list of predicted classes, for example [header, data, data].*

After each prompt, we provided the text of the input file, up to the maximum context length of each model, reserving 32 and 256 tokens for dialect detection and row classification output, respectively. A selection of the input files and responses can be seen in Figure 5 and Figure 6.

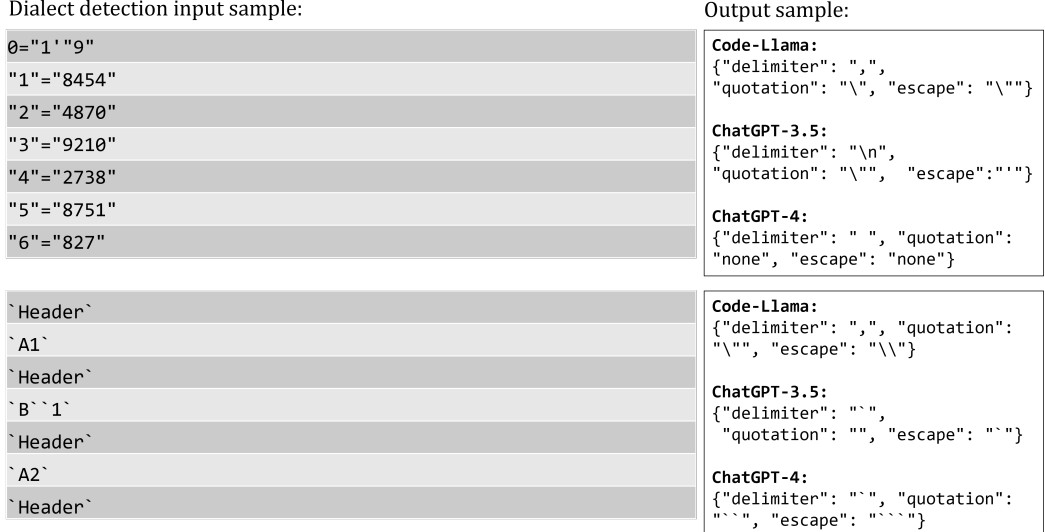

Figure 5: A sample of results asking LLMs to detect the dialect of challenging CSV files.

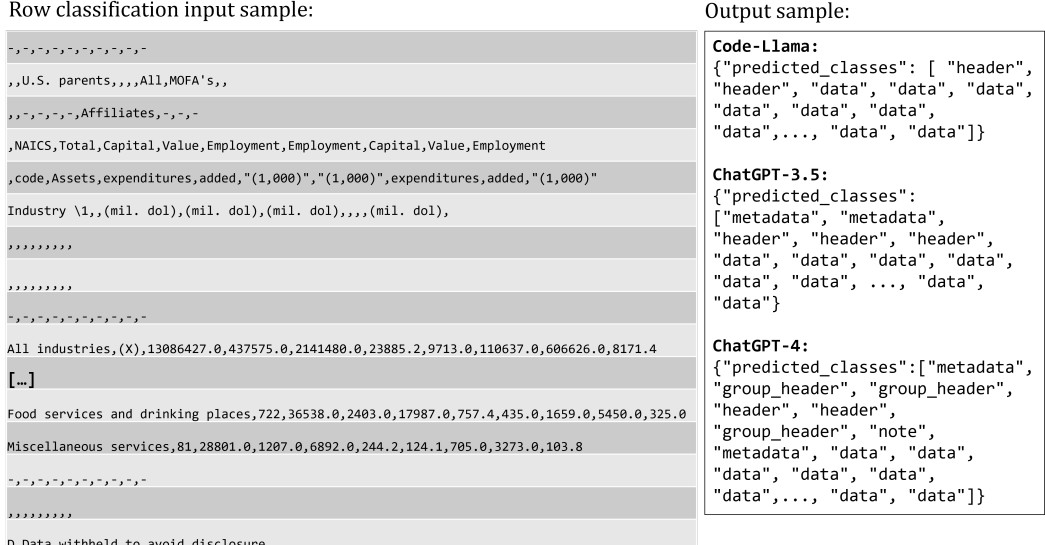

Figure 6: A sample result asking LLMs to detect the row classes for a challenging tabular file.

Beyond the poor experimental performances outlined in Sections 3 and 4, our experience dissuaded us from pursuing this approach for several reasons:

**Input considerations:** The models do not demonstrate real structural understanding of tabular files. In the example shown in Figure 5, when asked to detect the dialect of the first file, CODE-LLAMA returns as a delimiter the comma character, although the file does not contain a single comma. Similarly, the models of the ChatGPT family return the newline and space characters as the file delimiters, although they are not present within any of the rows.

**Output considerations:** The responses of the model, even if in JSON-like format, it is tuned towards natural language, and therefore be ambiguous and require parsing rules themselves. For example, in Figure 5, in one instance the CHATGPT-4 model replied with "none" as being a delimiter, while for the second file, null characters are indicated with an empty string. Moreover, we often had to infer customized parsing rules and repairs for poorly formatted JSON files, especially due to broken escape symbols and quotation characters.

**Generalizability considerations:** Given the highly stochastic nature of LLM, and the fact that they have been trained on a massive set of data, two considerations hinder a serious experimental evaluation of their generalizability performances. First, the files used for experimenting have possibly already been seen by these models, being publicly available at the time of the training of these models. Second, models may very well "hallucinate" outside the given prompt (Ji et al., 2023), and provide outputs completely unrelated to the task or outright wrong. In some instances, for the row classification task, the models picked up on one of the classes and kept repeating it until the given maximum response length was reached.

**Repeatability considerations:** Out of the model we tested, the better results were obtained with proprietary LLMs of the GPT family (OpenAI, 2023). These models are proprietary and closed-source, accessed through API calls. This limits any repeatability for experiments since there is no guarantee that, in the future, the internals of a model will not change (as they already did since the first experiments in our research), or that the models themselves will still be available. On the other hand, open source architectures like CODE-LLAMA (Touvron et al., 2023) still require significant hardware resources to reproduce their performances, e.g., parallel GPU computation with huge memory requirements.

All of these considerations make LLMs unreliable and difficult to integrate into a data management pipeline. Therefore, we resolved to pursue the design of a specialized framework for structural preparation: one that generalizes well with unseen files and is not sensitive to their content; not designed for natural language input/output but rather to be integrated with automated data management components; and that can be run on commonly available hardware.

# B  DATASET STATISTICS

**Pretraining:**  To pretrain the overall RENEMB architecture towards structural embedding, we use the raw files from the GitTables dataset (Hulsebos et al., 2023), which consists of 871 394 publicly available tabular files from GitHub. We note that, although these files are marked with the CSV extension, they do not necessarily conform to the RFC standard (Shafranovich, 2005), and may generally have metadata lines along with tables, non-standard dialects, or inconsistencies between rows. We believe these features make the pretraining of RENEMB more robust towards different types of tabular files. From each of the GitTables files, we sample 2 500 rows from each file (sampling with repetition for files having fewer rows than that). Then, we create a balanced dataset of 10 million row pairs, half of which are pairs extracted from the same file and the other half from two different randomly chosen files. The same dataset of pairs is used for the Structural Masking Modeling task and the Same File Prediction task since the transformer layers are trained on both tasks at the same time. To simplify pretraining complexity, as suggested by Devlin et al. (2019), we first pre-train the structural transformer for 15 epochs using a sequence length of 32 and complete the pretraining with 3 epochs using a sequence length of 128. In both cases, we used a batch size of 64, as it was the highest dimension fitting in memory while performing the full model training. Regarding the maximum length of the input file rows, after exploratory analysis of our training data, we set the number of input tokens for every row to 128, to balance complexity as coverage, as this length covers over 90% of the 10M input rows, with the rest having a significantly higher number of tokens.

| Most common dialects | | | | | Least common dialects | | | |
|---|---|---|---|---|---|---|---|---|
| Del. | Quo. | Esc. | #Files | | Del. | Quo. | Esc. | #Files |
| , | $\varepsilon$ | $\varepsilon$ | 2895 | | = | ” | $\varepsilon$ | 1 |
| , | ” | $\varepsilon$ | 2613 | | = | $\varepsilon$ | $\varepsilon$ | 1 |
| ; | $\varepsilon$ | $\varepsilon$ | 448 | ... | — | ’ | $\varepsilon$ | 1 |
| ; | ” | $\varepsilon$ | 437 | | — | ‘ | $\varepsilon$ | 1 |
| , | ” | ” | 346 | | , | — | $\varepsilon$ | 1 |

Table 3: The 5 most and least common dialects in the original dialect detection dataset. The character $\varepsilon$ denotes an empty character.

**Dialect detection:** The original dialect detection set from van den Burg et al. (2019) contains 7 235 CSV files with a very imbalanced distribution of dialects towards the most common dialects. Table 3 shows the 5 most common and the 5 least common dialects found in the dataset files, together with their frequency. The dataset has a long tail of different dialects, with 32 unique dialects, 11 unique delimiters, 5 unique quotation characters, and 4 unique escape characters. To compensate for this bias, we augment the files to obtain a balanced dataset. Each file of the dataset is augmented by taking its original cell values, and replacing the delimiter, quotation, and escape character with all valid combinations (183 in total) of the distinct dialect characters found in the original dataset. A valid dialect is one where the delimiter is different from the quotation character, and there is no escape character if file cells are not enclosed in quotation marks. We only augment files that do not contain any target dialect character in their content, to avoid generating invalid CSV files. Augmenting a file to have the empty delimiter $\varepsilon$ corresponds to generating single-column files. To do so for otherwise multi-column files, we remove all columns except for the first. To augment files towards dialects with quotation or escape characters, we include at least one delimiter or escaped quotation character within random cells of the files (with a probability of 5%), and quote cell values if needed, accordingly.

To perform the augmentation, first, we split the original dataset into a train/dev and a separate test fold, containing respectively 5 690 and 1 545 files. Then, we augment separately the files within these folds with all valid and applicable dialects. For the train/dev fold, we sample 100 files per dialect class, and then randomly sample 80% of the files for training and 20% of the files for validation purposes. For the test fold, we sample 20 files per dialect class. Considering a total of 183 dialect classes, the final set of files is therefore composed of 14 640 training files, 3 660 development files, and 3 660 hold-out testing files. Overall, the benefit of this augmentation scheme is not only that it counters class imbalance, but it also provides the model with files having the same cell contents but with different dialects. We believe this helps the generalization power of the model and prevents the model from overfitting on file content rather than on structure.

| | GOVUK | SAUS | CIUS | DEEX | MENDELEY | TROY |
|---|---|---|---|---|---|---|
| #files | 226 | 223 | 269 | 444 | 62 | 200 |
| #rows | 97 212 | 11 598 | 34 556 | 77 852 | 195 598 | 4 348 |
| #cols | 3482 | 3955 | 3656 | 6390 | 797 | 2282 |
| *header* | 519 | 576 | 435 | 1 222 | 86 | 280 |
| *data* | 93 584 | 9 469 | 31 845 | 74 245 | 194 786 | 2 898 |
| *group* | 850 | 283 | 119 | 302 | 27 | 42 |
| *derived* | 665 | 280 | 449 | 664 | 9 | 239 |
| *metadata* | 878 | 472 | 1 034 | 713 | 604 | 315 |
| *note* | 716 | 667 | 674 | 706 | 86 | 575 |
| #files with annotated column types | 110 | 153 | 264 | 260 | 12 | 66 |
| #column types | 32 | 26 | 21 | 66 | 11 | 19 |

Table 4: Row classification and column type detection dataset overview, with instances of row classes and column types.

**Row classification and column type annotation:** For the row classification and column type detection task, we use six datasets of real-world tabular files: GOVUK, SAUS, CIUS, DEEX, MENDELEY, and TROY. The first is composed of CSV files publicly shared on the UK governmental data

portal and introduced by Jiang et al. (2021). The second and third are datasets from the US Census department, introduced first by Gol et al. (2019), and extracted from the Statistical Abstract of the US and the Crime In the US report (hence the acronyms). The DEEX dataset was collected as part of the DeExcelerator project and contains annotated spreadsheets from the ENRON, FUSE, and EU-SES corpora (Eberius et al., 2013). The MENDELEY dataset, introduced by Jiang et al. (2021), and used for out-of-domain testing, is composed of publicly available files containing scientific research data from the online portal Mendeley Data. Finally, the TROY dataset, from George Nagy (2016), contains tables available on international statistical websites.

We note how none of the original datasets contained column type annotations, which we include in our public repository. To annotate column types from these six datasets we followed the same strategy used by Zhang et al. (2020) and Sun et al. (2023): we annotate columns with a semantic type from DBpedia Auer et al. (2007) if their headers (disambiguated with respect to spaces) match with the name of a DBpedia ontology or property. We annotate one column for each file and discard files for which no column can be matched with a DBpedia attribute.

