# OpenReview forum: "Embedding File Structure for Tabular File Preparation"
_ICLR.cc/2024/Conference — Submitted to ICLR 2024_

### Official Review · Reviewer_ffWZ · 2023-10-31

**Soundness:** 2 fair
**Presentation:** 2 fair
**Contribution:** 2 fair
**Rating:** 3
**Confidence:** 3

**Summary:**

This paper introduces a method RENEMB, which employs transformers and CNNs to extract structural information from sentences. They conduct experiments on dialect detection, table understanding, and column type annotation tasks to demonstrate the effectiveness of RENEMB.

**Strengths:**

1. The paper introduces RENEMB, which uniquely combines transformers and CNNs, to extract structural information from sentences.
2. RENEMB's effectiveness is demonstrated across multiple tasks - including dialect detection, table understanding, and column type annotation.

**Weaknesses:**

1. The problem definition in this paper is unclear. While the objective is to better represent the structural information of tabular files, a clear problem definition is absent. The final experiments are conducted on dialect detection, table understanding, and column type annotation tasks. What is the relationship between these three tasks and the paper's objective? Why were these specific tasks chosen?
2. The motivation is unreliable. The paper claims that previous methods primarily focused on semantic improvements while neglecting textual structural information. However, language models like BERT and GPT, during their pre-training phase, learn both semantic knowledge and structural information. Given this, is there still value in training a separate model solely for recognizing text structural information?
3. The paper asserts that "In solving the structural masking task, RENEMB has to learn the difference between special characters that belong within a cell (e.g., a comma delimiting the digits of a number) and those with a structural role (e.g., a comma as a cell delimiter)." However, the Structural Masking Modeling task simply instructs the model on where to output specific symbols; the model cannot differentiate the same symbol's different meanings based on its position.
4. The pre-training task "Same File Prediction" is not clearly described. Firstly, how is this logistic regression classifier obtained? Furthermore, the process of training BERT through this classifier isn't elaborated upon. Additionally, is the task of classifying whether two rows come from the same file reasonable? Different files might have rows with the same structure.
5. The paper lacks ablation studies. As a result, it's unclear how the two pre-training tasks and the CNN structure individually impact the final outcomes.
6. The paper presents a limited number of baseline methods, and they are relatively outdated(between 2019 and 2021). Additionally, the paper lacks analytical experiments to substantiate that the proposed method has learned superior textual structural information.

**Questions:**

Please see the comments in the weakness part.

---

> ### Author Response · Authors · 2023-11-16
> **Answering reviewer concerns, revised paper version**
>
> Thank you for the valuable review and comments. We answer the questions raised and point out to changes to our paper to address the review (major changes are highlighted in blue in the revised paper)
>
> - **Problem definition (W1)**: We now include a formal problem definition in Section 1 to motivate the choice of the three experimental tasks to demonstrate how our contributions provide a valid solution. Namely, our goal is to automate the data preparation of tabular files. The three experimental tasks are chosen as they constitute the most relevant tasks required to parse tabular data from raw files.
>
> - **Motivation (W2)**: Our motivation for the design of a novel architecture is the fact that large pre-trained language models (and their tokenizers) are typically trained to understand natural language, not textual data. As our revised experiments show, they are unfit to solve structural preparation tasks. Moreover, they are typically too large to be fine-tuned with datasets with few annotations, such as the ones that characterize data preparation tasks.
>
> - **Strucural modeling (W3)**: To better clarify this example statement, we now use a slightly different and more detailed formulation. As an intuition, consider that the masked modeling task instructs the model to output the correct symbol in the correct position. For the mentioned example (masking a comma within a quoted field), to reconstruct the correct character the model needs to understand several relevant structural aspects: fields of the row are delimited with a comma; fields are quoted if they contain delimiters; the position of the masked token is inside a field, and not between fields. In doing so, it learns to model which one is a “delimiter” comma, and which one is a part of field values.
>
> - **Same File Prediction (W4)**: We now describe the Same File Prediction head in more detail in Section 2. The head is a logistic regression classifier taking as input the embedding for the [CLS] token, and outputting a value in [0,1] signifying the probability that the two rows are from the same file. The model is then trained with backpropagation of the binary cross-entropy loss. As we also mention in the paper, we believe that the challenging nature of the task is precisely the reason why the transformer layers are forced to learn “meaningful” structure.
>
> - **Ablation study (W5)**: We did not include further experiments on our architecture for lack of paper space. If the reviewer feels that an ablation study would further improve our paper, we can include an experimental baseline without the use of CNN layers/pre-training as an appendix.
>
> - **Experimental baselines (W6)**: We now included more baselines for our experiments, namely ChatGPT 3.5, ChatGPT 4, and Code–Llama. Regarding the other non-LLM baselines chosen, to the best of our knowledge, they constitute the SOTA to address the tasks of dialect detection, row classification, and column type detection.

---

### Official Review · Reviewer_Wgj4 · 2023-10-31

**Soundness:** 2 fair
**Presentation:** 3 good
**Contribution:** 1 poor
**Rating:** 3
**Confidence:** 3

**Summary:**

This work introduces a transformer-based model framework designed to embed complex tables in a structurally aware manner. The process begins with pattern tokenization, followed by the use of structurally aware modules to capture the unique structural features of tables. The authors pre-train this framework on a dataset of 1M tabular files and evaluate its performance against state-of-the-art baselines.

However, the paper's contributions seem not so solid to me. The practice of employing structural features to encode structured text is not novel, as evidenced by prior research such as [1], and [2]. The authors' approach of utilizing a human-designed structural pattern tokenization to understand structures, followed by the use of basic modules for encoding, lacks technical novelty.

Moreover, recent debates have emerged regarding the optimal way to encode structured text: a structurally aware approach, or grounding and understanding structured text directly using text language models. The latter method has demonstrated superior performance and versatility across various structural forms, including tables, SQLs, and code, as shown in studies by [3] [4].

References:

[1] Li, Yulin, et al. "Structext: Structured text understanding with multi-modal transformers." Proceedings of the 29th ACM International Conference on Multimedia. 2021.

[2] Nassar, Ahmed, et al. "Tableformer: Table structure understanding with transformers." Proceedings of the IEEE/CVF Conference on Computer Vision and Pattern Recognition. 2022.

[3] Roziere, Baptiste, et al. "Code llama: Open foundation models for code." arXiv preprint arXiv:2308.12950 (2023).

[4] Xie, Tianbao, et al. "Unifiedskg: Unifying and multi-tasking structured knowledge grounding with text-to-text language models." arXiv preprint arXiv:2201.05966 (2022).

**Strengths:**

1. The paper is written with clarity, and is easy to understand.
2. RenEMB shows competitive performance with baselines on three diverse tasks.
3. RenEMB provides unique solutions to a few stated problems with table representations, including dialect detection, and structural awareness. These experiments show that RenEMB exhibit good structural representation abilities.

**Weaknesses:**

1. My major concern is on technical novelty and whether the method is up-to-date and competitive with SOTA table understanding generalist LLMs, as stated in the summary.

2. On pattern encoding method: In Section 2.1, I believe the patterns are not applicable to a multilingual setting. For instance, there are no upper/lower letters in Chinese or Korean language. Also, the pattern would be the same for tables with shared headings over a few columns, as the format of your table 1, and one that does not share headings.

3. On experiment settings: I am not sure if dialect detection and row classification constitute challenging tasks for table understanding. I believe more downstream tasks and finetuning analysis would be necessary. For example, what is the performance of tasks on TableQA
(WikiSQL, WikiTQ).

**Questions:**

Q1: What is the advantage of RenEMB over methods that directly tokenize tables with bpe and encode the tokenized text as transformers, a practice commonly used in training LLMs, as in Llama, GPT-3, Galactica?

Q2: Can this method apply to multilingual setting or could be used as a general method accross different kinds of table formats and languages?

---

> ### Author Response · Authors · 2023-11-16
> **Answer to the concerns expressed, proposed paper changes**
>
> Thank you for the detailed review and comments.  We group the discussion of individual concerns, and point out to revisions in the paper following the points identified in the review (major changes are in blue in the revised version of the paper).
>
> - **Novelty and contributions (W1)**: To address the concerns introduced in W1 regarding technical novelty, in the revised draft of the paper the related work section discusses in more detail how the structural representations learned with our approach differ from those of the mentioned [1-4]. In the introduction section we better clarify what are the novel contributions of our work. Furthermore, we now include a full experimental analysis for the performances of the SOTA LLM systems ChatGPT 3.5, ChatGPT 4, and CodeLlama on the file preparation tasks.
> Regarding novelty, we believe that our work shares the intuition of previous works to propose a specialized framework for understanding structured text, but is unique in that it is the first to apply to understanding the structure of tabular files. The proposed approaches in [1] and [2] are designed for document images (e.g., PDF documents) and apply either OCR methods or CNNs to identify the regions of the image that contain tables, then use transformers on their content to encode their semantic. In contrast, we employ a transformer architecture to learn a representation of the textual structure of data files, and a CNN architecture to represent the structure of the feature maps produced by the transformers - not on image data.
> The Code-Llama model proposed in [3] leverages an existing LLM pre-trained on natural language, and further specializes the training with the task of code infilling. With that work, we share the intuition of having a specialized pre-training stage, but our method pre-trains a transformer architecture from scratch with different training objectives, a different input domain, and different target tasks.
> Regarding the work in [4], we deem it complementary to ours: it proposes a framework to leverage structured information for semantic tasks, but its prerequisite is the correct parsing and integration of structured information from raw files: precisely the goal of our research.
>
> - **Pattern tokenization (Q1)**: The advantage of the RenEMB tokenization strategy lies in the fine-grained tokenization of special characters. Tokenization strategies used in SOTA language models (e.g., WordPiece or SentencePiece) are trained on natural language sentences, and therefore often overlook or combine special characters. For example, we identified errors in dialect detection for the GPT and Llama models where the identified delimiter was `","`. We attribute this error to tokenizers combining consecutive special characters in a single token, making it impossible to separate the end quotation, delimiter, and following begin quotation in a CSV file.
>
> - **Multi-language support (W2,Q2)**: Yes, the pattern abstraction method can be applied for multi-language support. In revising the paper, we included a sentence that clarifies this behavior.
> To clarify the specific doubts about the support of pictogram languages such as Chinese or Korean, our pattern tokenization strategy was designed referring to the Unicode symbol reference classes that include support for all languages, including pictogram-based ones. For example, the Chinese character 好 (UTF U+597D) for the word ‘hao’, belongs to the “Letter, other” class. This Unicode class is mapped to a ‘lowercase letter’ pattern. Beyond specific examples, we believe that referring to Unicode character classes is sufficient for all practical purposes in covering the distinction between the characters that encode values of a table, and the special characters used to delimit its structure.
>
> - **Semantic experiments (W3)**: The intended scope of RenEMB is the automation of file preparation tasks, which are often overlooked yet constitute significant time in typical data-driven workflows. As our model is not pre-trained for semantic tasks, we would not see the value of experimenting with traditional table or SQL question-answering tasks. We remain open to suggestion of further data preparation tasks for further experimental analysis.

---

### Official Review · Reviewer_9VNy · 2023-11-01

**Soundness:** 3 good
**Presentation:** 3 good
**Contribution:** 2 fair
**Rating:** 5
**Confidence:** 3

**Summary:**

The paper proposes a model named RenEmb to embed the structure of the tabular data.

There are 3 components of the model, namely pattern tokenization, then structural transformer, and CNN.

**Strengths:**

The paper seems well written and clear. The target problem seems to have broad interest and very useful importance. The approach seems feasible to be used by many researchers.

**Weaknesses:**

The major concern of the paper could seem:

a) The model seems only BERT-style and there seems no comparison to real LLM's such as chatgpt. The comparison to LLM's should be a core result, as many researchers have observed the level of paradigm shift by those new LLM.

b) The 1st step of pattern tokenization could make use of structure data that exclude the text content. Would it be enhanced if we have both structure and text? For the new hybrid approach of RenEmb + Strudel, can you let us know the reason that two concatenated models seem more advanced than just 1 stage?

**Questions:**

Would the new method work for complex table, not full row and full column?

---

> ### Author Response · Authors · 2023-11-16
> **Answering review questions and revised paper**
>
> Thank you for the valuable points and acknowledging the importance and usefulness of our research. We carefully considered the concerns expressed, and revised the paper (major changes are highlighted in blue in the revised paper).
>
> - **Experimental comparison**: To address the concerns raised in Point a), we have updated the paper with experimental comparison with ChatGPT 3.5, ChatGPT 4, and Code-Llama. The results of the experiments indicate that these LLMs show marginal capabilities to address the preparation tasks, but they cannot be leveraged reliably as they lack accuracy and true understanding of the structure of tabular data files.
>
> - **Tokenization**: To answer the first question of point b), we chose to focus on structure for the design of the tokenization, since after experimenting with architectures that use word piece/semantic tokenizers (see the experimental performances for the Roberta model), we had no success in training them to effectively solve our tasks. We attribute the lack of success to the fact that, if too much semantic content is left in the input, attention layers “overfit” on learning semantic dependencies between values and don’t focus on structural aspects.
>
> - **Ensembling**: Regarding the second question in Point b), we now include a paragraph in Section 4 to clarify the effect of the combination of our approach with Strudel.
>
> - **Complex inputs**: The answer to the last question is “yes”: thanks to the convolutional file embedding stage, our model is capable of processing complex tabular files that deviate from a “typical” standard CSV format. In fact, the datasets used for the experiments of Sections 4 and 5 contain files with these complex structures, e.g., metadata rows and multiple tables within a single file.

---

### Meta-Review · Area_Chair_7Q66 · 2023-12-20

**Metareview:**

This paper proposes a method RENEMB, which employs transformers and CNNs to represent schema information from text.

Strengths:
1. The paper proposes a model RENEMB based on Transformer and CNNs to extract structural information.
2. The paper demonstrates RENEMS's performance on dialect detection, table understanding, and column type annotation.

Weaknesses:
1. All reviewers share one major concern about the paper: it does not demonstrate the effectiveness of the proposed approach on downstreaming tasks with and without the file structural (or schema) information. The downstream tasks are not the file structure understanding itself, but understanding tasks such as question answering for the file with and without schema embedding.

Reviewer 9VNy's first weakness comment is already addressed by the author's new revision. Reviewer Wgj4's comment on four related works is also addressed by the authors. The works on document understanding are not closely related on this task. They are for different problems. Reviewer ffWZ's comments 1-4 are not well addressed by the authors.

**Justification For Why Not Higher Score:**

Major weaknesses are unresolved.

**Justification For Why Not Lower Score:**

N/A

---

### Decision · Program_Chairs · 2024-01-16

Reject